# Analyzing Hidden Representations in End-to-End Automatic Speech Recognition Systems

**Yonatan Belinkov** and **James Glass**
Computer Science and Artificial Intelligence Laboratory
Massachusetts Institute of Technology
Cambridge, MA 02139
`{belinkov, glass}@mit.edu`

## Abstract

Neural networks have become ubiquitous in automatic speech recognition systems. While neural networks are typically used as acoustic models in more complex systems, recent studies have explored end-to-end speech recognition systems based on neural networks, which can be trained to directly predict text from input acoustic features. Although such systems are conceptually elegant and simpler than traditional systems, it is less obvious how to interpret the trained models.

In this work, we analyze the speech representations learned by a deep end-to-end model that is based on convolutional and recurrent layers, and trained with a connectionist temporal classification (CTC) loss. We use a pre-trained model to generate frame-level features which are given to a classifier that is trained on frame classification into phones. We evaluate representations from different layers of the deep model and compare their quality for predicting phone labels. Our experiments shed light on important aspects of the end-to-end model such as layer depth, model complexity, and other design choices.

## 1  Introduction

Traditional automatic speech recognition (ASR) systems are composed of multiple components, including an acoustic model, a language model, a lexicon, and possibly other components. Each of these is trained independently and combined during decoding. As such, the system is not directly trained on the speech recognition task from start to end. In contrast, end-to-end ASR systems aim to map acoustic features directly to text (words or characters). Such models have recently become popular in the ASR community thanks to their simple and elegant architecture [1, 2, 3, 4]. Given sufficient training data, they also perform fairly well. Importantly, such models do not receive explicit phonetic supervision, in contrast to traditional systems that typically rely on an acoustic model trained to predict phonetic units (e.g. HMM phone states). Intuitively, though, end-to-end models have to generate some internal representation that allows them to abstract over phonological units. For instance, a model that needs to generate the word "bought" should learn that in this case "g" is not pronounced as the phoneme /g/.

In this work, we investigate if and to what extent end-to-end models *implicitly* learn phonetic representations. The hypothesis is that such models need to create and exploit internal representations that correspond to phonetic units in order to perform well on the speech recognition task. Given a pre-trained end-to-end ASR system, we use it to extract frame-level features from an acoustic signal. For example, these may be the hidden representations of a recurrent neural network (RNN) in the end-to-end system. We then feed these features to a classifier that is trained to predict a phonetic property of interest such as phone recognition. Finally, we evaluate the performance of the classifier as a measure of the quality of the input features, and by proxy the quality of the original ASR system.

We aim to provide quantitative answers to the following questions:

1. To what extent do end-to-end ASR systems learn phonetic information?
2. Which components of the system capture more phonetic information?
3. Do more complicated models learn better representations for phonology? And is ASR performance correlated with the quality of the learned representations?

Two main types of end-to-end models for speech recognition have been proposed in the literature: connectionist temporal classification (CTC) [1, 2] and sequence-to-sequence learning (seq2seq) [3, 4]. We focus here on CTC and leave exploration of the seq2seq model for future work.

We use a phoneme-segmented dataset for the phoneme recognition task, as it comes with time segmentation, which allows for accurate mapping between speech frames and phone labels. We define a frame classification task, where given representations from the CTC model, we need to classify each frame into a corresponding phone label. More complicated tasks can be conceived of—for example predicting a single phone given all of its aligned frames—but classifying frames is a basic and important task to start with.

Our experiments reveal that the lowest layers in a deep end-to-end model are best suited for representing phonetic information. Applying one convolution on input features improves the representation, but a second convolution greatly degrades phone classification accuracy. Subsequent recurrent layers initially improve the quality of the representations. However, after a certain recurrent layer performance again drops, indicating that the top layers do not preserve all the phonetic information coming from the bottom layers. Finally, we cluster frame representations from different layers in the deep model and visualize them in 2D, observing different quality of grouping in different layers.

We hope that our results would promote the development of better ASR systems. For example, understanding representation learning at different layers of the end-to-end model can guide joint learning of phoneme recognition and ASR, as recently proposed in a multi-task learning framework [5].

## 2 Related Work

### 2.1 End-to-end ASR

End-to-end models for ASR have become increasingly popular in recent years. Important studies include models based on connectionist temporal classification (CTC) [1, 2, 6, 7] and attention-based sequence-to-sequence models [3, 4, 8]. The CTC model is based on a recurrent neural network that takes acoustic features as input and is trained to predict a symbol per each frame. Symbols are typically characters, in addition to a special blank symbol. The CTC loss then marginalizes over all possible sequences of symbols given a transcription. The sequence-to-sequence approach, on the other hand, first encodes the sequence of acoustic features into a single vector and then decodes that vector into the sequence of symbols (characters). The attention mechanism improves upon this method by conditioning on a different summary of the input sequence at each decoding step.

Both these of these approaches to end-to-end ASR usually predict a sequence of characters, although there have also been initial attempts at directly predicting words [9, 10].

### 2.2 Analysis of neural representations

While end-to-end neural network models offer an elegant and relatively simple architecture, they are often thought to be opaque and uninterpretable. Thus researchers have started investigating what such models learn during the training process. For instance, previous work evaluated neural network acoustic models on phoneme recognition using different acoustic features [11] or investigated how such models learn invariant representations [12] and encode linguistic features [13, 14]. Others have correlated activations of gated recurrent networks with phoneme boundaries in autoencoders [15] and in a text-to-speech system [16]. Recent work analyzed different speaker representations [17]. A joint audio-visual model of speech and lip movements was developed in [18], where phoneme embeddings were shown to be closer to certain linguistic features than embeddings based on audio alone. Other joint audio-visual models have also analyzed the learned representations in different ways [19, 20, 21]. Finally, we note that analyzing neural representations has also attracted attention in other domains

Table 1: The ASR models used in this work.

(a) DeepSpeech2.

| Layer | Type | Input Size | Output Size |
|---|---|---|---|
| 1 | cnn1 | 161 | 1952 |
| 2 | cnn2 | 1952 | 1312 |
| 3 | rnn1 | 1312 | 1760 |
| 4 | rnn2 | 1760 | 1760 |
| 5 | rnn3 | 1760 | 1760 |
| 6 | rnn4 | 1760 | 1760 |
| 7 | rnn5 | 1760 | 1760 |
| 8 | rnn6 | 1760 | 1760 |
| 9 | rnn7 | 1760 | 1760 |
| 10 | fc | 1760 | 29 |

(b) DeepSpeech2-light.

| Layer | Type | Input Size | Output Size |
|---|---|---|---|
| 1 | cnn1 | 161 | 1952 |
| 2 | cnn2 | 1952 | 1312 |
| 3 | lstm1 | 1312 | 600 |
| 4 | lstm2 | 600 | 600 |
| 5 | lstm3 | 600 | 600 |
| 6 | lstm4 | 600 | 600 |
| 7 | lstm5 | 600 | 600 |
| 8 | fc | 600 | 29 |

like vision and natural language processing, including word and sentence representations [22, 23, 24], machine translation [25, 26], and joint vision-language models [27]. To our knowledge, hidden representations in end-to-end ASR systems have not been thoroughly analyzed before.

# 3 Methodology

We follow the following procedure for evaluating representations in end-to-end ASR models. First, we train an ASR system on a corpus of transcribed speech and freeze its parameters. Then, we use the pre-trained ASR model to extract frame-level feature representations on a phonemically transcribed corpus. Finally, we train a supervised classifier using the features coming from the ASR system, and evaluate classification performance on a held-out set. In this manner, we obtain a quantitative measure of the quality of the representations that were learned by the end-to-end ASR model. A similar procedure has been previously applied to analyze a DNN-HMM phoneme recognition system [14] as well as text representations in neural machine translation models [25, 26].

More formally, let $\mathbf{x}$ denote a sequence of acoustic features such as a spectrogram of frequency magnitudes. Let $\texttt{ASR}_t(\mathbf{x})$ denote the output of the ASR model at the $t$-th input. Given a corresponding label sequence, $\mathbf{l}$, we feed $\texttt{ASR}_t(\mathbf{x})$ to a supervised classifier that is trained to predict a corresponding label, $l_t$. In the simplest case, we have a label at each frame and perform frame classification. As we are interested in analyzing different components of the ASR model, we also extract features from different layers $k$, such that $\texttt{ASR}_t^k(\mathbf{x})$ denotes the output of the $k$-th layer at the $t$-th input frame.

We next describe the ASR model and the supervised classifier in more detail.

## 3.1 ASR model

The end-to-end model we use in this work is DeepSpeech2 [7], an acoustics-to-characters system based on a deep neural network. The input to the model is a sequence of audio spectrograms (frequency magnitudes), obtained with a 20ms Hamming window and a stride of 10ms. With a sampling rate of 16kHz, we have 161 dimensional input features. Table 1a details the different layers in this model. The first two layers are convolutions where the number of output feature maps is 32 at each layer. The kernel sizes of the first and second convolutional layers are 41x11 and 21x11 respectively, where a convolution of TxF has a size T in the time domain and F in the frequency domain. Both convolutional layers have a stride of 2 in the time domain while the first layer also has a stride of 2 in the frequency domain. This setting results in 1952/1312 features per time frame after the first/second convolutional layers.

The convolutional layers are followed by 7 bidirectional recurrent layers, each with a hidden state size of 1760 dimensions. Notably, these are simple RNNs and not gated units such as long short-term memory networks (LSTM) [28], as this was found to produce better performance. We also consider a simpler version of the model, called DeepSpeech2-light, which has 5 layers of bidirectional LSTMs, each with 600 dimensions (Table 1b). This model runs faster but leads to worse recognition results.

Each convolutional or recurrent layer is followed by batch normalization [29, 30] and a ReLU non-linearity. The final layer is a fully-connected layer that maps onto the number of symbols (29 symbols: 26 English letters plus space, apostrophe, and a blank symbol).

The network is trained with a CTC loss [31]:

$$L = -\log p(\mathbf{l}|\mathbf{x})$$

where the probability of a label sequence $\mathbf{l}$ given an input sequence $\mathbf{x}$ is defined as:

$$p(\mathbf{l}|\mathbf{x}) = \sum_{\pi \in \mathcal{B}^{-1}(\mathbf{l})} p(\pi|\mathbf{x}) = \sum_{\pi \in \mathcal{B}^{-1}(\mathbf{l})} \prod_{t=1}^{T} \mathtt{ASR}_t^K(\mathbf{x})[\pi_t]$$

where $\mathcal{B}$ removes blanks and repeated symbols, $\mathcal{B}^{-1}$ is its inverse image, $T$ is the length of the label sequence $\mathbf{l}$, and $\mathtt{ASR}_t^K(\mathbf{x})[j]$ is unit $j$ of the model output after the top softmax layer at time $t$, interpreted as the probability of observing label $j$ at time $t$. This formulation allows mapping long frame sequences to short character sequences by marginalizing over all possible sequences containing blanks and duplicates.

## 3.2   Supervised Classifier

The frame classifier takes features from different layers of the DeepSpeech2 model as input and predicts a phone label. The size of the input to the classifier thus depends on which layer in DeepSpeech2 is used to generate features. We model the classifier as a feed-forward neural network with one hidden layer, where the size of the hidden layer is set to 500.[1] This is followed by dropout (rate of 0.5) and a ReLU non-linearity, then a softmax layer mapping onto the label set size (the number of unique phones). We chose this simple formulation as we are interested in evaluating the quality of the representations learned by the ASR model, rather than improving the state-of-the-art on the supervised task.

We train the classifier with Adam [32] with the recommended parameters ($\alpha = 0.001$, $\beta_1 = 0.9$, $\beta_2 = 0.999$, $\epsilon = e^{-8}$) to minimize the cross-entropy loss. We use a batch size of 16, train the model for 30 epochs, and choose the model with the best development loss for evaluation.

## 4   Tools and Data

We use the `deepspeech.torch` [33] implementation of Baidu's DeepSpeech2 model [7], which comes with pre-trained models of both DeepSpeech2 and the simpler variant DeepSpeech2-light. The end-to-end models are trained on LibriSpeech [34], a publicly available corpus of English read speech, containing 1,000 hours sampled at 16kHz. The word error rates (WER) of the DeepSpeech2 and DeepSpeech2-light models on the Librispeech-test-clean dataset are 12 and 15, respectively [33].

For the phoneme recognition task, we use TIMIT, which comes with time segmentation of phones. We use the official train/development/test split and extract frames for the frame classification task. Table 2 summarizes statistics of the frame classification dataset. Note that due to sub-sampling at the DeepSpeech2 convolutional layers, the number of frames decreases by a factor of two after each convolutional layer. The possible labels are the 60 phone symbols included in TIMIT (excluding the begin/end silence symbol *h#*). We also experimented with the reduced set of 48 phones used by [35].

The code for all of our experiments is publicly available.[2]

Table 2: Frame classification data extracted from TIMIT.

|                       | Train   | Development | Test   |
|-----------------------|---------|-------------|--------|
| Utterances            | 3,696   | 400         | 192    |
| Frames (input)        | 988,012 | 107,620     | 50,380 |
| Frames (after cnn1)   | 493,983 | 53,821      | 25,205 |
| Frames (after cnn2)   | 233,916 | 25,469      | 11,894 |

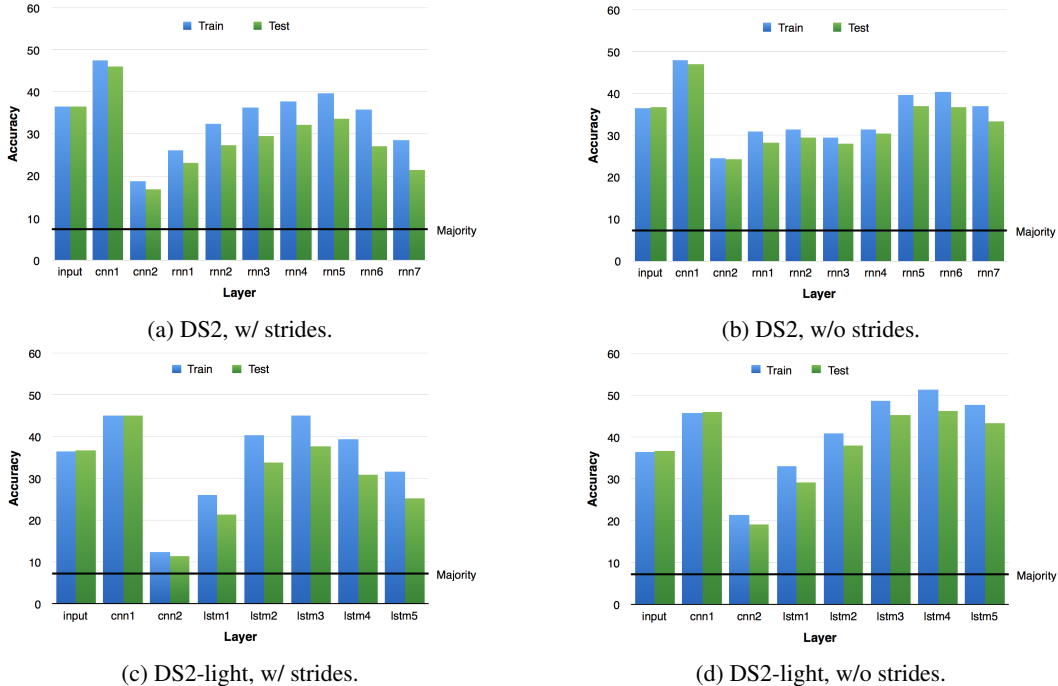

| | |
|---|---|
| (a) DS2, w/ strides. | (b) DS2, w/o strides. |
| (c) DS2-light, w/ strides. | (d) DS2-light, w/o strides. |

Figure 1: Frame classification accuracy using representations from different layers of DeepSpeech2 (DS2) and DeepSpeech2-light (DS2-light), with or without strides in the convolutional layers.

## 5 Results

Figure 1a shows frame classification accuracy using features from different layers of the DeepSpeech2 model. The results are all above a majority baseline of 7.25% (the phone "s"). Input features (spectrograms) lead to fairly good performance, considering the 60-wise classification task. The first convolution further improves the results, in line with previous findings about convolutions as feature extractors before recurrent layers [36]. However, applying a second convolution significantly degrades accuracy. This can be attributed to the filter width and stride, which may extend across phone boundaries. Nevertheless, we find the large drop quite surprising.

The first few recurrent layers improve the results, but after the 5th recurrent layer accuracy goes down again. One possible explanation to this may be that higher layers in the model are more sensitive to long distance information that is needed for the speech recognition task, whereas the local information that is needed for classifying phones is better captured in lower layers. For instance, to predict a word like "bought", the model would need to model relations between different characters, which would be better captured at the top layers. In contrast, feed-forward neural networks trained on phoneme recognition were shown to learn increasingly better representations at higher layers [13, 14]; such networks do not need to model the full speech recognition task, different from end-to-end models.

In the following sections, we first investigate three aspects of the model: model complexity, effect of strides in the convolutional layers, and effect of blanks. Then we visualize frame representations in 2D and consider classification into abstract sound classes. Finally, Appendix A provides additional experiments with windows of input features and a reduced phone set, all exhibiting similar trends.

### 5.1 Model complexity

Figure 1c shows the results of using features from the DeepSpeech2-light model. This model has less recurrent layers (5 vs. 7) and smaller hidden states (600 vs. 1760), but it uses LSTMs instead of simple RNNs. A first observation is that the overall trend is the same as in DeepSpeech2: significant drop after the first convolutional layer, then initial increase followed by a drop in the final layers.

Comparing the two models (figures 1a and 1c), a number of additional observations can be made. First, the convolutional layers of DeepSpeech2 contain more phonetic information than those of

DeepSpeech2-light (+1% and +4% for cnn1 and cnn2, respectively). In contrast, the recurrent layers in DeepSpeech2-light are better, with the best result of 37.77% in DeepSpeech2-light (by lstm3) compared to 33.67% in DeepSpeech2 (by rnn5). This suggests again that higher layers do not model phonology very well; when there are more recurrent layers, the convolutional layers compensate and generate better representations for phonology than when there are fewer recurrent layers. Interestingly, the deeper model performs better on the speech recognition task while its deep representations are not as good at capturing phonology, suggesting that its top layers focus more on modeling character sequences, while its lower layers focus on representing phonetic information.

## 5.2 Effect of strides

The original DeepSpeech2 models have convolutions with strides (steps) in the time dimension [7]. This leads to subsampling by a factor of 2 at each convolutional layer, resulting in reduced dataset size (Table 2). Consequently, the comparison between layers before and after convolutions is not entirely fair. To investigate this effect, we ran the trained convolutions without strides during feature generation for the classifier.

Figure 1b shows the results at different layers without using strides in the convolutions. The general trend is similar to the strided case: large drop at the 2nd convolutional layer, then steady increase in the recurrent layers with a drop at the final layers. However, the overall shape of the accuracy in the recurrent layers is less spiky; the initial drop is milder and performance does not degrade as much at the top layers. A similar pattern is observed in the non-strided case of DeepSpeech2-light (Figure 1d).

These results can be attributed to two factors. First, running convolutions without strides maintains the number of examples available to the classifier, which means a larger training set. More importantly, however, the time resolution remains high which can be important for frame classification.

## 5.3 Effect of blank symbols

Recall that the CTC model predicts either a letter in the alphabet, a space, or a blank symbol. This allows the model to concentrate probability mass on a few frames that are aligned to the output symbols in a series of spikes, separated by blank predictions [31]. To investigate the effect of blank symbols on phonetic representation, we generate predictions of all symbols using the CTC model, including blanks and repetitions. Then we break down the classifier's performance into cases where the model predicted a blank, a space, or another letter.

Figure 2 shows the results using representations from the best recurrent layers in DeepSpeech2 and DeepSpeech2-light, run with and without strides in the convolutional layers. In the strided case, the hidden representations are of highest quality for phone classification when the model predicts a blank. This appears counterintuitive, considering the spiky behavior of CTC models, which should be more confident when predicting non-blank. However, we found that only 5% of the frames are predicted as blanks, due to downsampling in the strided convolutions. When the model is run without strides, we observe a somewhat different behavior. Note that in this case the model predicts many more blanks (more than 50% compared to 5% in the non-strided case), and representations of frames predicted as blanks are not as good, which is more in line with the common spiky behavior of CTC models [31].

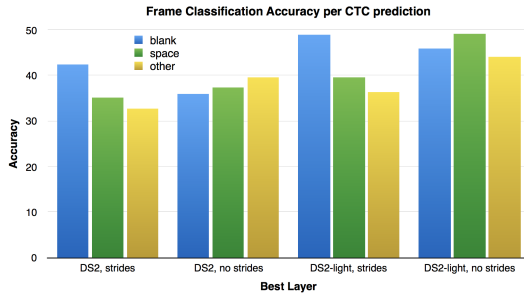

Figure 2: Frame classification accuracy at frames predicted as blank, space, or another letter by DeepSpeech2 and DeepSpeech2-light, with and without strides in the convolutional layers.

## 5.4 Clustering and visualizing representations

In this section, we visualize frame representations from different layers of DeepSpeech2. We first ran the DeepSpeech2 model on the entire development set of TIMIT and extracted feature representations for every frame from all layers. This results in more than 100K vectors of different sizes (we use the model without strides in convolutional layers to allow for comparable analysis across layers). We followed a similar procedure to that of [20]: We clustered the vectors in each layer with k-means ($k = 500$) and plotted the cluster centroids using t-SNE [37]. We assigned to each cluster the phone label that had the largest number of examples in the cluster. As some clusters are quite noisy, we also consider pruning clusters where the majority label does not cover enough of the cluster members.

Figure 3 shows t-SNE plots of cluster centroids from selected layers, with color and shape coding for the phone labels (see Figure 9 in Appendix B for other layers). The input layer produces clusters which show a fairly clean separation into groups of centroids with the same assigned phone. After the input layer it is less easy to detect groups, and lower layers do not show a clear structure. In layers rnn4 and rnn5 we again see some meaningful groupings (e.g. "z" on the right side of the rnn5 plot), after which rnn6 and rnn7 again show less structure.

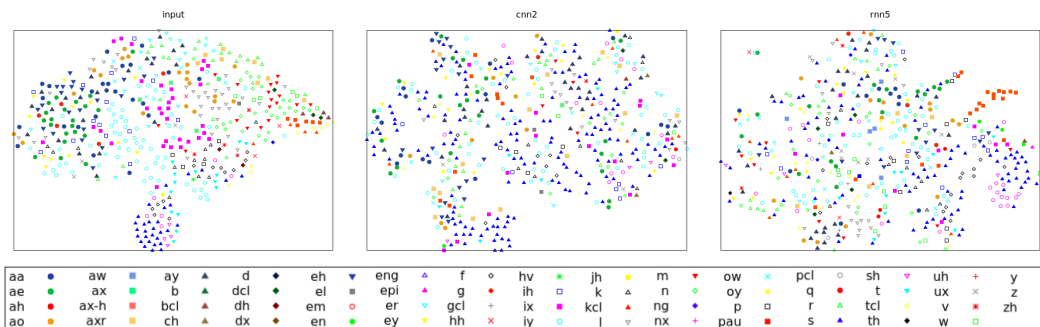

Figure 3: Centroids of frame representation clusters using features from different layers.

Figure 10 (in Appendix B) shows clusters that have a majority label of at least 10-20% of the examples (depending on the number of examples left in each cluster after pruning). In this case groupings are more observable in all layers, and especially in layer rnn5.

We note that these observations are mostly in line with our previous findings regarding the quality of representations from different layers. When frame representations are better separated in vector space, the classifier does a better job at classifying frames into their phone labels; see also [14] for a similar observation.

## 5.5 Sound classes

Speech sounds are often organized in coarse categories like consonants and vowels. In this section, we investigate whether the ASR model learns such categories. The primary question we ask is: which parts of the model capture most information about coarse categories? Are higher layer representations more informative for this kind of abstraction above phones? To answer this, we map phones to their corresponding classes: affricates, fricatives, nasals, semivowels/glides, stops, and vowels. Then we train classifiers to predict sound classes given representations from different layers of the ASR model.

Figure 4 shows the results. All layers produce representations that contain a non-trivial amount of information about sound classes (above the vowel majority baseline). As expected, predicting sound classes is easier than predicting phones, as evidenced by a much higher accuracy compared to our previous results. As in previous experiments, the lower layers of the network (input and cnn1) produce the best representations for predicting sound classes. Performance then first drops at cnn2 and increases steadily with each recurrent layer, finally decreasing at the last recurrent layer. It appears that higher layers do not generate better representations for abstract sound classes.

Next we analyze the difference between the input layer and the best recurrent layer (rnn5), broken down to specific sound classes. We calculate the change in F1 score (harmonic mean of precision and recall) when moving from input representations to rnn5 representations, where F1 is calculated in two

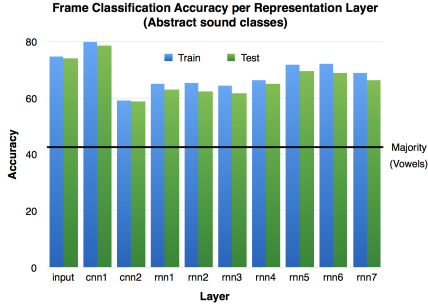

Figure 4: Accuracy of classification into sound classes using representations from different layers of DeepSpeech2.

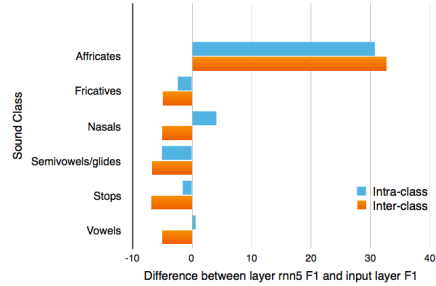

Figure 5: Difference in F1 score using representations from layer rnn5 compared to the input layer.

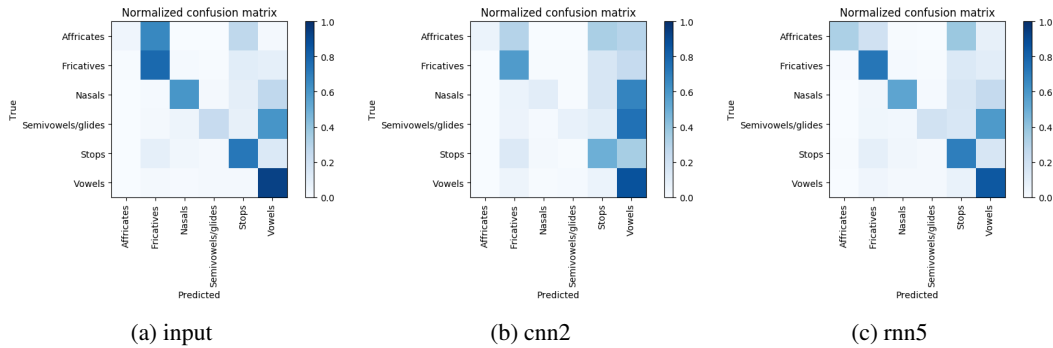

(a) input  (b) cnn2  (c) rnn5

Figure 6: Confusion matrices of sound class classification using representations from different layers.

ways. The *inter-class* F1 is calculated by directly predicting coarse sound classes, thus measuring how often the model confuses two separate sound classes. The *intra-class* F1 is obtained by predicting fine-grained phones and micro-averaging F1 inside each coarse sound class (not counting confusion outside the class). It indicates how often the model confuses different phones in the same sound class.

As Figure 5 shows, in most cases representations from rnn5 degrade the performance, both within and across classes. There are two notable exceptions. Affricates are better predicted at the higher layer, both compared to other sound classes and when predicting individual affricates. It may be that more contextual information is needed in order to detect a complex sound like an affricate. Second, the intra-class F1 for nasals improves with representations from rnn5, whereas the inter-class F1 goes down, suggesting that rnn5 is better at distinguishing between different nasals.

Finally, Figure 6 shows confusion matrices of predicting sound classes using representations from the input, cnn2, and rnn5 layers. Much of the confusion arises from confusing relatively similar classes: semivowels/vowels, affricates/stops, affricates/fricatives. Interestingly, affricates are less confused at layer rnn5 than in lower layers, which is consistent with our previous observation.

## 6 Conclusion

In this work, we analyzed representations in a deep end-to-end ASR model that is trained with a CTC loss. We empirically evaluated the quality of the representations on a frame classification task, where each frame is classified into its corresponding phone label. We compared feature representations from different layers of the ASR model and observed striking differences in their quality. We also found that these differences are partly correlated with the separability of the representations in vector space.

In future work, we would like to extend this analysis to other speech features, such as speaker and dialect ID, and to larger speech recognition datasets. We are also interested in experimenting with other end-to-end systems, such as sequence-to-sequence models and acoustics-to-words systems. Another venue for future work is to improve the end-to-end model based on our insights, for example by improving the representation capacity of certain layers in the deep neural network.

## Acknowledgements

We would like to thank members of the MIT spoken language systems group for helpful discussions. This work was supported by the Qatar Computing Research Institute (QCRI).

## Footnotes

[1]We also experimented with a linear classifier and found that it produces lower results overall but leads to similar trends when comparing features from different layers.

[2]`http://github.com/boknilev/asr-repr-analysis`

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
