[Supplementary Material]

# Analyzing Hidden Representations in End-to-End Automatic Speech Recognition Systems Supplementary Material

**Yonatan Belinkov** and **James Glass**
Computer Science and Artificial Intelligence Laboratory
Massachusetts Institute of Technology
Cambridge, MA 02139
{belinkov, glass}@mit.edu

## A  Additional experiments

### A.1  Windows of features

Our main experiments used a simple frame representation by taking the output of the ASR model at frame $t$, $\text{ASR}_t(\mathbf{x})$. We also consider a window of features around the frame at time $t$. This improves the representation and also accounts for possible delay effects [1]. Test set results with different window widths are shown in Figure 7 (DeepSpeech model, no strides). As expected, larger windows improve the representation quality. The absolute numbers are much better than using only a single frame (+10-15%), but the overall trend for a given window size is similar: initial performance drop after the convolutional layers, then steady increase at the first recurrent layers and another drop at the top layers. The drop is somewhat more moderate than in the single frame case (compare to Figure 1b), indicating that some shifting effect may indeed be taking place, although it might be limited given that we are using bidirectional RNNs (the results in [1] are with unidirectional RNNs).

Figure 7: Frame classification accuracy using different window widths around the current frame.

## A.2 Reduced phone set

In addition to the full set of 60 phones and the coarse sound classes, we also experimented with a reduced set of 48 phones [2]. As Figure 8 shows, the trend is similar to the other phone sets. We also noticed, as with sound classes (Section 5.5), that the affricates /ch/ and /jh/ are better represented at rnn5 (F1 score of 42.5% and 34.9%, respectively) than at the input layer (7.2% and 8.3%).

Figure 8: Frame classification accuracy with a reduced set of 48 phones.

## B   Visualizations of frame representations

Figure 9 shows a t-SNE [3] visualization of cluster centroids of activations from different layers in DeepSpeech2, where each cluster is assigned the phone label that had the largest number of examples in the cluster. The input layer produces clusters which show a fairly clean separation into groups of centroids with the same assigned phone. After the input layer it is less easy to detect groups, and lower layers do not show a clear structure. In layers rnn4 and rnn5 we again see some meaningful groupings, after which rnn6 and rnn7 again show less structure.

Figure 10 shows clusters that have a majority label of at least 10-20% of the examples (depending on the number of examples left in each cluster after pruning). In this case groupings are more observable in all layers, and especially in layer rnn5.

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

Figure 9: Centroids of all frame representation clusters using features from different layers.

Figure 10: Centroids of frame representation clusters using features from different layers, showing only clusters where the majority label covers at least 10-20% of the cluster members.