[Reviews · NeurIPS 2017]

Reviewer 1



The paper examines the extent to which a character-based CTC system (an “End to End” system) uses phonemes as an internal representation. To answer this question the hidden activations of the network are classified using either a feed forward neural network or a nearest neighbour classifier. Visualizations and through analysis are presented. The paper is clearly written and the methodology is sound. I believe it will be of interest for the speech community. I especially like the analysis in section 5.5 which sheds some light onto classification of phonemes with longer temporal structured. Other than that I have the following constructive comments: - Section 4: The baseline ASR system performance should be reported on both LibriSpeech and on TIMIT, to assure the reader the task is carried out on a mature system. - The message of section 5.3 is not clear to me. There are small trends all over the place, none are really explained. - Line 207: “...hidden states carry information from past observations.” This is not accurate as the system at hand is bidirectional and also the future is observed. - Line 210: It's not clear what is “the original CTC model”? - Section 5.4: Figure 5.3 is too small and dense, you can unify legend for all plots and remove values from the axis.

Reviewer 2



This paper tries to analyze the speech representations learned by a deep end-to-end model. I have the following comments: 1. The paper uses a phone classification task to evaluate the quality of the speech representation. But phone classification cannot directly be translated into tasks like automatic speech recognition. It is only one aspect of the speech representation, and I don't think it's a good idea to use it as the primary metric of the representation quality. 2. It is not super clear what the authors learned from the analysis, and how those points can be used to improve actual speech related tasks such as automatic speech recognition. 3. Figure 1 shows the classification accuracy using features from different layers of the neural network. Does it imply that the phone classification accuracy may not be a good metric for speech representation?

Reviewer 3



The authors conduct an analysis of CTC trained acoustic models to determine how information related to phonetic categories is preserved in CTC-based models which directly output graphemes. The work follows a long line of research that has analyzed neural network representations to determine how they model phonemic representations, although to the best of my knowledge this has not been done previously for CTC-based end-to-end architectures. The results and analysis presented by the authors is interesting, although there are some concerns I have with the conclusions that the authors draw that I would like to clarify these points. Please see my detailed comments below. - In analyzing the quality of features learned at each layer in the network, from the description of the experiments in the paper it would appear that the authors only feed in features corresponding to a single frame at time t, and predict the corresponding output label. In the paper, the authors conclude that (Line 159--164) "... after the 5th recurrent layer accuracy goes down again. One possible explanation to this may be that higher layers in the model are more sensitive to long distance information that is needed for the speech recognition task, whereas the local information which is needed for classifying phones is better captured in lower layers." This is a plausible explanation, although another plausible explanation that I'd like to suggest for this result is the following: It is know from previous work, e.g., (Senior et al., 2015) that even when predicting phoneme output targets, CTC-based models with recurrent layers can significantly delay outputs from the model. In other words, the features at a given frame may not contain information required to predict the current ground-truth label, but this may be shifted. Thus, for example, it is possible that if a window of features around the frame at time t was used instead of a single frame, the conclusions vis-a-vis the quality of the recurrent layers in the model would be very different. An alternative approach, would be to follow the procedure in (Senior et al., 2015) and constrain the CTC loss function to only allow a certain amount of delay in predicting labels. I think it is important to conduct the experiments required to establish whether this is the case before drawing the conclusions in the paper. Reference: A. Senior, H. Sak, F. de Chaumont Quitry, T. Sainath and K. Rao, "Acoustic modelling with CD-CTC-SMBR LSTM RNNS," 2015 IEEE Workshop on Automatic Speech Recognition and Understanding (ASRU), Scottsdale, AZ, 2015, pp. 604-609. - In analyzing phoneme categories I notice that the authors use the full 61 label TIMIT phoneme set (60 after excluding h#). However, it is common practice for TIMIT to also report phoneme recognition results after mapping the 61 label set down to 48 following (Lee and Hon, 89). In particular, this maps certain allophones into the same category, e.g., many closures are mapped to the same class. It would be interesting to confirm that the same trends as identified by the authors continue to hold in the reduced set. Reference: Lee, K-F., and H-W. Hon. "Speaker-independent phone recognition using hidden Markov models." IEEE Transactions on Acoustics, Speech, and Signal Processing 37.11 (1989): 1641-1648. - There have been many previous works on analyzing neural network representations in the context of ASR apart from the references cited by the authors. I would suggest that the authors incorporate some of the references listed below in addition to the ones that they have already listed: * Nagamine, Tasha, Michael L. Seltzer, and Nima Mesgarani. "On the Role of Nonlinear Transformations in Deep Neural Network Acoustic Models." INTERSPEECH. 2016. * Nagamine, Tasha, Michael L. Seltzer, and Nima Mesgarani. "Exploring how deep neural networks form phonemic categories." Sixteenth Annual Conference of the International Speech Communication Association. 2015. * Yu, Dong, et al. "Feature learning in deep neural networks-studies on speech recognition tasks." arXiv preprint arXiv:1301.3605 (2013). - In the context of grapheme-based CTC for ASR, I think the authors should cite the following early work: * F. Eyben, M. Wöllmer, B. Schuller and A. Graves, "From speech to letters - using a novel neural network architecture for grapheme based ASR," 2009 IEEE Workshop on Automatic Speech Recognition & Understanding, Merano, 2009, pp. 376-380. - Line 250: "... and micro-averaging F1 inside each coarse sound class." What is meant by "micro-averaging" in this context? - Minor comments and typographical errors: * Line 135: "We train the classifier with Adam [22] with default parameters ...". Please specify what you mean by default parameters. * In Figure 1 and Figure 2: The authors use "steps" when describing strides in the convolution. I'd recommend changing this to "strides" for consistency with the text. * In Figure 1. d: The scale on the Y-axis is different from the other figures which makes the two figures in (c.) and (d.) not directly comparable. Could the authors please use the same Y-axis scale for all figures in Figure 1.